

# Cloninger's TCI associations with adaptive and maladaptive emotion regulation strategies

Han Chae[1], Soo Hyun Park[2], Danilo Garcia[3,4] and Soo Jin Lee[5]

[1] School of Korean Medicine, Pusan National University, Busan, South Korea
[2] Department of Psychology, Yonsei University, Seoul, South Korea
[3] Blekinge Center of Competence, Region Blekinge, Karlskrona, Sweden
[4] Department of Psychology, University of Gothenburg, Gothenburg, Sweden
[5] Department of Psychology, Kyungsung University, Busan, South Korea

## ABSTRACT

**Background**. Cognitive emotion regulation plays a crucial role in psychopathology, resilience and well-being by regulating response to stress situations. However, the relationship between personality and adaptive and maladaptive regulation has not been sufficiently examined.

**Methods**. Adaptive and maladaptive cognitive emotion regulation strategies of 247 university students were measured using the Cognitive Emotion Regulation Questionnaire (CERQ) and their temperament and character characteristics were analyzed with the Temperament and Character Inventory—Revised Short (TCI-RS). Two-step hierarchical multiple regression analyses were used to analyze whether TCI-RS explains the use of adaptive and maladaptive cognitive emotion regulation strategies. The latent classes of cognitive emotion regulation strategies were extracted with Latent Class Analysis (LCA) and significant differences in the subscales of CERQ and TCI-RS were examined with Analysis of Covariance (ANCOVA) and Profile Analysis after controlling for sex and age.

**Results**. The two-step hierarchical multiple regression model using the seven TCI-RS subscales explained 32.30% of the adaptive and 41.70% of the maladaptive CERQ subscale scores when sex and age were introduced in the first step as covariates. As for temperament, Novelty Seeking (NS) and Persistence (PS) were pivotal for adaptive and Harm Avoidance (HA) and PS for maladaptive CERQ total scores. In addition, the character traits Self-Directedness (SD) and Cooperativeness (CO) were critical for high adaptive and low maladaptive CERQ scores. Four latent emotion regulation classes were confirmed through LCA, and distinct TCI-RS profiles were found. The temperament trait HA and character trait SD were significantly different among the four latent emotion regulation classes.

**Discussion**. This study demonstrated that SD and CO are related to cognitive emotion regulation strategies along with psychological health and well-being, and that PS exhibits dualistic effects when combined with NS or HA on response to stressful situations. The importance of developing mature character represented by higher SD and CO in regard to mental health and its clinical implementation was discussed.

Corresponding author
Soo Jin Lee, leesooj@ks.ac.kr, leesooj@gmail.com

## INTRODUCTION

We deal with a variety of situations in everyday life, which leads to various emotional responses. Such emotional responses appear immediate and unchangeable at first. However, biological, social, and cognitive behavioral self-regulatory processes become activated internally. These processes are responsible for monitoring, evaluating and modifying emotional reactions in order to accomplish one's goals in the context of human life, and such a process has been termed emotion regulation (*Garnefski, Kraaij & Spinhoven, 2001*; *Thompson, 1994*). Successful emotion regulation is associated with optimal physical and mental health outcomes (*Aldao, Nolen-Hoeksema & Schweizer, 2010*; *Jo & Lee, 2013*; *Potthoff et al., 2016*; *Seol & Park, 2015*), improved relationships (*Cicchetti, Ackerman & Izard, 1995*; *Mihalca & Tarnavska, 2013*) and better academic and work performance (*Grandey & Melloy, 2017*; *Mega, Ronconi & De Beni, 2014*). Difficulties in emotion regulation are associated with mental disorders including major depression, bipolar disorder, generalized anxiety disorder, social anxiety disorder, eating disorder, and alcohol and substance related problems (*Aldao, Nolen-Hoeksema & Schweizer, 2010*).

Two separate strategies can be broadly defined as (1) mature, adaptive and positive vs. (2) immature, maladaptive and negative, and they are the product of predisposed tendencies based on individual differences (*Garnefski, Kraaij & Spinhoven, 2001*).

These two types of strategies, adaptive (adaptive cognitive emotion regulation questionnaire: aCERQ) and maladaptive (maladaptive cognitive emotion regulation questionnaire: mCERQ), can be measured using the CERQ (*Garnefski, Kraaij & Spinhoven, 2001*). The CERQ consists of nine subscales corresponding to adaptive or maladaptive strategies. The aCERQ consists of five subscales: Putting into perspective (PIP), Refocus on planning (REP), Positive Refocusing (PRF), Positive Reappraisal (PRA) and Acceptance (ACC). The mCERQ is composed of four subscales: Rumination (RUM), Catastrophizing (CAT), Blaming Others (BLO) and Blaming Self (BLS).

In this context, personality has been suggested to predict the use of emotion regulation strategies of a given person (*Kokkonen & Pulkkinen, 2001*). Previous studies using temperamental models of personality, for example, have reported that those high in Neuroticism tend to use maladaptive emotion regulation strategies, while individuals high in Extraversion and Openness use adaptive emotion regulation strategies (*Hassani et al., 2008*; *Kokkonen & Pulkkinen, 2001*; *Purnamaningsih, 2017*; *Sadr, 2016*; *Sohn & Yoo, 2011*). However, personality also consists of psychosocial characteristics within an individual that determine behavioral, emotional and cognitive responses to internal and environmental stimuli that extend beyond temperament or stable automatic emotional responses (*Cloninger, 2004*). Thus, given the fact that successful emotion regulation in the face of stressful stimuli is an adaptive and mature coping strategy, we argue that what predisposes it is a stable temperament or the tendency to demonstrate stable emotional

reactions (e.g., low scores in Neuroticism). Furthermore, mature character is reflected by changeable and mature goals and values in relation to the self, others, and the whole world (*Eley et al., 2016*; *Kim, Lee & Lee, 2013*; *Moreira et al., 2015*).

Cloninger's biopsychosocial model of personality (*Cloninger, 1987*; *Cloninger, Svrakic & Przybeck, 1993*) consists of two domains, temperament and character, and explains the foundation and development of personality through their interaction. Temperament is an innate and biological foundation for character (*Gray, 1982*; *Sjobring, 1973*). In other words, it constrains character development but does not fully determine it because of the systematic effects of sociocultural learning and the stochastic effects of experience (*Cloninger, 1987*; *Cloninger, Svrakic & Przybeck, 1993*; *Cloninger & Zwir, 2018*; *Garcia et al., 2014*).

The temperament domain is composed of NS, HA, Reward Dependence (RD), and PS. NS is a personality trait associated with exploratory excitability in response to novel stimuli, and HA is associated with behavioral inhibition in reaction to dangerous or abhorrent stimuli. RD is a tendency to respond particularly to signals of social reward, and PS is a tendency to maintain behavior with intermittent reinforcement (*Cloninger, Svrakic & Przybeck, 1993*). The character domain is comprised of SD, CO, and Self-Transcendence (ST). SD indicates the tendency to identify the self as an autonomous individual and CO measures the degree to which one considers the self as integral to humanity. ST represents the degree to which one considers the self as an integral part of the universe as a whole (*Cloninger, Svrakic & Przybeck, 1993*). Generally speaking, there are advantages of both high and low temperament scores depending on the situations but from the perspective of character, only high scores are adaptive.

In addition, higher score in SD is a critical protective factor in psychopathology and also a factor in the promotion of mental health (*Eley et al., 2013*; *Eley et al., 2016*; *Kim, Lee & Lee, 2013*). Meanwhile, higher scores in HA are positively associated to maladaptive emotion regulation as measured by CERQ (*Aldao, Nolen-Hoeksema & Schweizer, 2010*; *Ball, Smolin & Shekhar, 2002*; *Izadpanah et al., 2016*; *Manfredi et al., 2011*; *Peirson & Heuchert, 2001*; *Schreiber, Grant & Odlaug, 2012*). The combination of higher scores in SD and lower scores in HA which reflect positive characteristics such as life satisfaction, optimism, and hopefulness (*Cloninger, 2004*) is important for mental health and longevity (*DeNeve & Cooper, 1998*; *Lee et al., 2014*).

However, previous studies were primarily dedicated to examining the strong association between maladaptive emotion regulation strategies, psychopathology and well-being (*Aldao, Nolen-Hoeksema & Schweizer, 2010*). Therefore, there is a need for research that considers positive and adaptive perspectives as well as negative and maladaptive perspectives because health is characterized by not only the absence of illness, but also a state of well-being (*WHO, 2005*). In addition, HA and SD traits along with other temperament and character traits should be examined together. For instance, resilience or a person's ability to effectively adapt and flourish in face of adversity is associated not only with low scores in HA and high scores in SD, but also with the ability to be hard working and persistent (i.e., high scores in PS) (*Eley et al., 2013*).

For these reasons, the aim of this study was to explore the association between personality and adaptive and maladaptive cognitive emotion regulation by (1) assessing how much variance in emotion regulation scores can be explained by personality scores and by (2) identifying groups of emotion regulation style and assessing differences in personality scores between these groups (*Cloninger et al., 2012*; *Eley et al., 2016*). This will help us to understand how temperament and character interact to impact adaptive and maladaptive emotional responses to stress, hence providing foundations for promoting psychological well-being.

## MATERIALS & METHODS

### Participants and procedures

The CERQ and TCI-RS were used to measure cognitive emotional regulation strategies and personality of 313 college students who enrolled in an elective course of a metropolitan university. A total of 247 (89 men and 158 women) out of 313 students chose to participate (78.9%) voluntarily after a researcher introduced and completed the questionnaires. The participants were also asked about their age, sex, and education level as the part of their survey. This study was approved by the institutional ethics board of Pusan National University (PNU IRB/2015_59_HR) and all of the participants provided the active and written informed consent.

### Cognitive Emotion Regulation Questionnaire (CERQ)

The CERQ measures the cognitive style or emotional regulation strategies when overcoming stress from negative life events and it has been frequently used to examine its relationship to psychopathology (*Garnefski, Kraaij & Spinhoven, 2001*). Emotion regulation plays an important role in managing biopsychosocial response to stress, and CERQ measures the use of specific response strategies from a cognitive perspective, rather than a behavioral and external perspective (*Garnefski, Kraaij & Spinhoven, 2001*).

CERQ is composed of 36 questions using a 5-point scale (0 = almost never to 4 = almost always). The CERQ consists of two scales, aCERQ and mCERQ, along with nine subscales. The aCERQ (20 items, response range = 0–80) is composed of five subscales, namely PIP (e.g., "I think that it all could have been much worse"), REP (e.g., "I think of what I can do best"), PRF (e.g., "I think of nicer things than what I have experienced"), PRA (e.g., "I think I can learn something from the situation") and ACC (e.g., "I think that I have to accept that this has happened"), and the mCERQ (16 items, response range = 0–64) consists of four subscales, namely RUM (e.g., "I often think about how I feel about what I have experienced"), CAT (e.g., "I keep thinking about how terrible it is what I have experienced"), BLO (e.g., "I feel that others are to blame for it") and BLS (e.g., "I feel that I am the one to blame for it").

The Korean version of the CERQ was validated by *Kim (2004)* with the same 36 items and 5-point scale based on the original questionnaire. The internal consistencies (Cronbach's $\alpha$) were reported as .66 for PIP, .80 for REP, .85 for PRF, .80 for PRA, .53 for ACC, .68 for RUM, .78 for CAT, .83 for BLO, and .76 for BLS (*Kim, 2004*). The internal consistencies

in our present study were as followings: .69 for PIP, .82 for REP, .88 for PRF, .82 for PRA, .55 for ACC, .74 for RUM, .79 for CAT, .85 for BLO, and .78 for BLS.

## Temperament and Character Inventory-Revised Short (TCI-RS)

The TCI, developed by *Cloninger, Svrakic & Przybeck (1993)*, consists of two interrelated domains of temperament and character. The four temperament traits reflect biases in automatic responses to emotional stimuli involving involuntary rational processes, whereas the three character traits represent differences in higher cognitive functions underlying one's goals, values, and relationships (*Cloninger, 2008*; *Cloninger, Svrakic & Przybeck, 1993*).

The temperament dimension is composed of HA (anticipatory worry, fear of uncertainty, shyness with strangers, and fatigability), NS (characterized by exploratory excitability, impulsiveness, extravagance, and disorderliness), RD (sentimentality, openness, attachment, and dependence), and PS (eagerness, work-hardened, ambition, and perfectionism) (*Chae et al., 2014*).

The character dimension consists of SD (characterized by purposeful, responsible, resourceful, and self-accepting when high scores in SD), CO (empathic, helpful, forgiving, and tolerant when high scores in CO), and ST (contemplating, idealistic, spiritual, and transpersonal when high scores in ST).

The Korean version of the Temperament and Character Inventory-Revised-Short (TCI-RS) is a 140-item self-report questionnaire validated for use in Korea based on the original TCI (*Min, Oh & Lee, 2007*). Respondents rate each item on a 5-point scale (0=definitely false to 4 = definitely true). The internal consistencies (Cronbach's $\alpha$) of HA, NS, RD, PS, SD, CO, and ST were shown as .86, .83, .81, .82, .87, .76, and .90, respectively (*Min, Oh & Lee, 2007*).

## Statistical analysis

The significant differences in age, school year, college major and subscales of CERQ and TCI-RS between male and female participants were examined using $\chi^2$ and $t$-tests. The correlations between subscales of CERQ and TCI-RS were examined with Pearson's correlation analysis. Two-step hierarchical multiple regression analyses were conducted to examine how TCI-RS temperament and character explains adaptive and maladaptive strategy scores on the CERQ. Sex and age were introduced in the first step as covariates (model 1) and all seven TCI-RS subscales were added in the second step (model 2).

Latent Class Analysis (LCA) was used to explore the latent classes of all nine subscales of CERQ using Mplus 5.21 (*Lee, Choi & Chae, 2017*; *Muthén & Muthén, 2005*). LCA is an empirically driven statistical model that reveals subgroups of individuals based on similar characteristics (*Lanza, Flaherty & Collins, 2003*) from observed variables, such as the five adaptive and four maladaptive strategy subscales of CERQ, and estimates the probability of any individual falling into a specific subgroup. The number of latent classes or subgroups is determined by comparison of fit statistics (see below). More specifically, LCA models are estimated with classes added repeatedly to determine which model demonstrates the best fit to the current data until the addition of classes no longer demonstrates significant improvement in model fit statistics.

The criteria of model fit were as follows (*Nylund, Asparouhov & Muthen, 2007*): (1) the smaller the Bayesian Information Criterion (BIC) and adjusted BIC, the better the model; (2) the more significant the probability values of Vuong-Lo-Mendell-Rubin Likelihood Difference Test (VLMR), Lo-Mendell-Rubin Likelihood Difference Test (LMR) or Bootstrap Likelihood Ratio Test (BLRT), the better the model; and (3) an Entropy index greater than .8, which reflects the distinctiveness of latent classes is assumed as good (*Muthén, 2004*). If the criteria fit more than one class, the interpretation of the researcher is important in selecting the number of classes.

After determining the number of latent classes, ANCOVA was performed to compare aCERQ and mCERQ total scores among extracted latent CERQ classes. Following ANCOVA, Bonferroni post-hoc analysis was performed to confirm where the differences among the classes occurred. Profile Analysis was also conducted to analyze significant differences in the TCI-RS subscales between the extracted latent CERQ classes with sex and age as covariates (*Lee, Choi & Chae, 2017*). To calculate flatness and parallelism, we incorporated the Greenhouse-Geisser correction when Mauchly's sphericity test was significant.

The data are presented as means and standard deviations or as frequencies with percentages, and estimated values for sex and age are shown as means and standard errors. All analysis was performed using IBM SPSS Statistics 20.0 (IBM, Armonk, NY) and Mplus 5.21 (Muthén & Muthén, Los Angeles, CA).

## RESULTS

### Demographic characteristics of the participants in this study

The demographic features of the participants are shown in Table 1. There were significant differences in age ($t = 4.92$, $p < .001$), but not in university year ($\chi^2 = 2.07$, $p = .569$) or college major ($\chi^2 = 18.95$, $p = .090$) between male and female participants. The higher age of male participants most likely resulted from the fact that male students in Korea have to take a two-year leave of absence during their university enrollment to complete their mandatory military service. Since demographic features showed distinctive sex and age differences, we performed all further statistical analyses with sex and age as covariates.

The male participants showed significantly higher aCERQ total ($t = 3.47$, $p < .001$), PIP ($t = 2.13$, $p < .05$), ROP ($t = 4.41$, $p < .001$), PRF ($t = 2.35$, $p < .05$) and PRA ($t = 2.75$, $p < .01$) scores than female participants. However, there was no significant difference in mCERQ ($t = -0.01$, $p = .998$). For the TCI-RS subscales, male participants demonstrated significantly higher PS ($t = 4.08$, $p < .001$) and SD ($t = 3.50$, $p < .001$) scores, while female participants showed significantly higher scores in HA ($t = -3.36$, $p < .001$) than male participants.

### Correlation between subscales of CERQ and TCI-RS

Correlations between CERQ and TCI-RS subscales were examined using Pearson's correlation analysis (Table 2). The aCERQ total and mCERQ total scores were found to be independent ($r = -.08$, $p = .206$). The aCERQ total score was significantly correlated with HA ($r = -.44$, $p < .001$), PS ($r = .47$, $p < .001$), SD ($r = .42$, $p < .001$) and CO

**Table 1  Demographic features of the participants.**

| | Male | Female | Total | $\chi^2$ or $t$ | $p$ |
|---|---|---|---|---|---|
| $n$ | 89 (36%) | 158 (64%) | 247 | | |
| **Age**[***] | 21.57 ± 2.20 | 20.29 ± 1.44 | | 4.92 | <.001 |
| Years in University | | | | 2.07 | .569 |
| 1 | 31 | 45 | 76 | | |
| 2 | 25 | 47 | 72 | | |
| 3 | 19 | 31 | 50 | | |
| 4 | 14 | 35 | 49 | | |
| Major | | | | 18.95 | .090 |
| Business | 4 | 8 | 12 | | |
| Economy & Trade | 8 | 18 | 26 | | |
| Engineering & Nano Tech. | 12 | 8 | 20 | | |
| Education | 7 | 8 | 15 | | |
| Sociology | 13 | 30 | 43 | | |
| Bio-resources | 5 | 4 | 9 | | |
| Environment | 1 | 8 | 9 | | |
| Arts | 1 | 11 | 12 | | |
| Humanities | 18 | 35 | 53 | | |
| Natural Science | 14 | 24 | 38 | | |
| Nursing/Dental | 6 | 4 | 10 | | |
| CERQ | | | | | |
| **aCERQ total**[***] | 53.72 ± 11.08 | 48.52 ± 11.42 | 50.39 ± 11.55 | 3.47 | **<.001** |
| **PIP**[*] | 10.58 ± 2.79 | 9.75 ± 3.05 | 10.05 ± 2.98 | 2.13 | **<.05** |
| **REP**[***] | 12.15 ± 2.60 | 10.51 ± 2.92 | 11.10 ± 2.91 | 4.41 | **<.001** |
| **PRF**[*] | 8.75 ± 4.06 | 7.53 ± 3.87 | 7.97 ± 3.98 | 2.35 | **<.05** |
| **PRA**[**] | 11.39 ± 3.13 | 10.14 ± 3.61 | 10.59 ± 3.49 | 2.75 | **<.01** |
| ACC | 10.84 ± 2.17 | 10.60 ± 2.43 | 10.69 ± 2.34 | 0.78 | .437 |
| mCERQ total | 28.09 ± 7.43 | 28.09 ± 9.81 | 28.09 ± 9.01 | −0.01 | .998 |
| RUM | 9.15 ± 2.89 | 9.12 ± 3.57 | 9.13 ± 3.33 | 0.06 | .951 |
| CAT | 4.98 ± 3.34 | 5.21 ± 3.73 | 5.13 ± 3.59 | −0.49 | .628 |
| BLO | 4.64 ± 2.79 | 5.13 ± 3.36 | 4.95 ± 3.17 | −1.16 | .248 |
| BLS | 9.33 ± 3.02 | 8.64 ± 3.21 | 8.89 ± 3.16 | 1.65 | .101 |
| TCI-RS | | | | | |
| NS | 35.61 ± 10.32 | 36.61 ± 10.98 | 36.25 ± 10.74 | −0.71 | .480 |
| **HA**[***] | 39.60 ± 12.97 | 45.31 ± 12.73 | 43.25 ± 13.08 | −3.36 | **<.001** |
| RD | 44.36 ± 11.21 | 46.65 ± 10.05 | 45.83 ± 10.52 | −1.65 | .100 |
| **PS**[***] | 46.74 ± 11.00 | 40.78 ± 11.03 | 42.93 ± 11.36 | 4.08 | **<.001** |
| **SD**[***] | 47.44 ± 11.51 | 41.92 ± 12.10 | 43.91 ± 12.16 | 3.50 | **<.001** |

**Table 1** (*continued*)

|  | Male | Female | Total | $\chi^2$ or $t$ | $p$ |
|---|---|---|---|---|---|
| CO | $57.56 \pm 9.34$ | $55.72 \pm 9.16$ | $56.38 \pm 9.25$ | 1.51 | .132 |
| ST | $21.46 \pm 10.00$ | $23.30 \pm 10.32$ | $22.64 \pm 10.22$ | $-1.36$ | .174 |

**Notes.**
    \*$p < .05$.
    \*\*$p < .01$.
    \*\*\*$p < .001$.
    CERQ, Cognitive Emotion Regulation Questionnaire; aCERQ, adaptive CERQ; mCERQ, maladaptive CERQ; TCI-RS, Temperament and Character Inventory-Revised Short; PIP, Putting into perspective; REP, Refocus on planning; PRF, Positive Refocusing; PRA, Positive Reappraisal; ACC, Acceptance; RUM, Rumination; CAT, Catastrophizing; BLO, Blaming Other; BLS, Blaming Self; NS, Novelty-Seeking; HA, Harm-Avoidance; RD, Reward-Dependence; PS, Persistence; SD, Self-Directedness; CO, Cooperativeness; ST, Self-Transcendence.

($r = .42$, $p < .001$), and the mCERQ total score was significantly correlated with HA ($r = .41$, $p < .001$) and SD ($r = -.42$, $p < .001$). For the TCI-RS subscales, HA showed a significant negative correlation with PS ($r = -.48$, $p < .001$) and SD ($r = -.76$, $p < .001$), and there was a significant positive correlation ($r = .57$, $p < .001$) between PS and SD.

## Regression analysis on aCERQ and mCERQ total scores with TCI-RS subscales, sex and age

A two-step hierarchical multiple regression analysis (Table 3) examined the significant degree of variance in temperament and character using sex and age as covariates by entering the covariates in the first step of the regression model.

The regression model [$F(3, 237) = 18.84$, $p < .001$, $R^2 = .42$ ($adj.R^2 = .40$)] explained 41.70% of the total variance in aCERQ total score, and it was found that age, NS, PS, SD, CO and ST were predictors of aCERQ total score. The regression model [$F(9, 237) = 12.59$, $p < .001$, $R^2 = .32$ ($adj.R^2 = .30$)] explained 32.30% of the variance in mCERQ total score, and it was found that HA, PS, SD and CO were predictors of mCERQ total score.

## LCA of CERQ subscale scores

LCA was performed (Table 4), and a four latent class model (Fig. 1A) was determined to be acceptable since all the criteria of model fit were satisfied: a four class model demonstrated smaller BIC (11212.21) and adjusted BIC (11053.71) compared to other models, more significant $p$ value of VLMR ($p = .009$), LMR ($p = .010$), and BLRT ($p = .000$), and a greater than .8 Entropy index (.82) (Table 4).

The four latent CERQ classes were labeled as HALM or High Adaptive and Low Maladaptive ($n = 78$, 32.58%), MALM or Middle Adaptive and Low Maladaptive ($n = 61$, 24.70%), MAHM or Middle Adaptive and High Maladaptive ($n = 91$, 36.84%) and LAHM or Low Adaptive and High Maladaptive ($n = 17$, 6.88%) classes based on aCERQ and mCERQ scores. aCERQ and mCERQ total score of HALM ($62.20 \pm .74$, $24.37 \pm .65$), MALM ($43.17 \pm .83$, $19.14 \pm .73$), LAHM ($29.17 \pm 1.56$, $37.59 \pm 1.38$), MAHM ($49.08 \pm .67$, $35.51 \pm .59$) classes were significantly different from each other according to ANCOVA ($F = 170.88$, $p < .001$ in aCERQ; $F = 128.84$, $p < .001$ in mCERQ).

Significant differences in all nine CERQ subscales among the four latent classes were found when ANCOVA and Bonferroni post-hoc analysis were performed. In aCERQ subscales, the four latent classes were significantly different from each other in regard

Peer J

**Table 2  Correlation coefficient among subscales of CERQ and TCI-RS.**

| | 1 | 2 | 3 | 4 | 5 | 6 | 7 | 8 | 9 | 10 | 11 | 12 | 13 | 14 | 15 | 16 | 17 | 18 |
|---|---|---|---|---|---|---|---|---|---|---|---|---|---|---|---|---|---|---|
| 1.aCERQ total score | 1 | .77*** | .65*** | .74*** | .84*** | .65*** | −.08 | .08 | −.29*** | −.12 | .13* | .09 | −.44*** | .25*** | .47*** | .42*** | .42*** | .29*** |
| 2.PIP | | 1 | .32*** | .47*** | .58*** | .46*** | −.10 | .02 | −.26*** | −.13* | .13* | .03 | −.33*** | .14* | .24*** | .27*** | .37*** | .23*** |
| 3.REP | | | 1 | .31*** | .43*** | .37*** | .06 | .12 | −.10 | .02 | .15* | .10 | −.31*** | .14* | .56*** | .41*** | .22*** | .17** |
| 4.PRF | | | | 1 | .50*** | .22*** | −.20** | −.04 | −.26*** | −.08 | −.14* | .00 | −.42*** | .24*** | .30*** | .36*** | .34*** | .21** |
| 5.PRA | | | | | 1 | .52*** | −.11 | .04 | −.29*** | −.18** | .14* | .06 | −.37*** | .20** | .38*** | .35*** | .35*** | .22*** |
| 6.ACC | | | | | | 1 | .15* | .24*** | −.08 | −.06 | .32*** | .17** | −.08 | .17** | .22** | .09 | .23*** | .24*** |
| 7.mCERQ total score | | | | | | | 1 | .75*** | .81*** | .60*** | .53*** | .18* | .41*** | .10 | −.00 | −.42*** | −.17** | .18** |
| 8.RUM | | | | | | | | 1 | .44*** | .23*** | .34*** | .16* | .25*** | .18** | .04 | −.29*** | −.03 | .23*** |
| 9.CAT | | | | | | | | | 1 | .50*** | .22** | .11 | .49*** | .01 | −.14* | −.47*** | −.25*** | .09 |
| 10.BLO | | | | | | | | | | 1 | −.09 | .19** | .22** | −.07 | −.02 | −.20** | −.31*** | .02 |
| 11.BLS | | | | | | | | | | | 1 | .03 | .13* | .14* | .12 | −.17** | .14* | .14* |
| 12.NS | | | | | | | | | | | | 1 | −.02 | .12 | .13* | −.20** | −.20** | .18** |
| 13.HA | | | | | | | | | | | | | 1 | −.12 | −.48*** | −.76*** | −.33*** | .01 |
| 14.RD | | | | | | | | | | | | | | 1 | .17** | .00 | .39*** | .27*** |
| 15.PS | | | | | | | | | | | | | | | 1 | .57*** | .33*** | .16* |
| 16.SD | | | | | | | | | | | | | | | | 1 | .31*** | −.07 |
| 17.CO | | | | | | | | | | | | | | | | | 1 | .25*** |
| 18.ST | | | | | | | | | | | | | | | | | | 1 |

**Notes.**
*$p < .05$
**$p < .01$
***$p < .001$
Bold represents correlation coefficient more than .4.
CERQ, Cognitive Emotion Regulation Questionnaire; aCERQ, adaptive CERQ; mCERQ, maladaptive CERQ; TCI-RS, Temperament and Character Inventory-Revised Short; PIP, Putting into perspective; REP, Refocus on planning; PRF, Positive Refocusing; PRA, Positive Reappraisal; ACC, Acceptance; RUM, Rumination; CAT, Catastrophizing; BLO, Blaming Other; BLS, Blaming Self; NS, Novelty-Seeking; HA, Harm-Avoidance; RD, Reward-Dependence; PS, Persistence; SD, Self-Directedness; CO, Cooperativeness; ST, Self-Transcendence.
**Table 3  Two-step hierarchical regression analysis on total score of aCERQ and mCERQ with seven subscales of TCI-RS, sex and age as covariates.**

| | Unstandardized | | Standardized | $t$ | $p$ |
|---|---|---|---|---|---|
| | **B** | **SE** | **beta** | | |
| aCERQ total score | | | | | |
| age[**] | **.97** | **.33** | **.16** | **2.90** | **.004** |
| sex | −1.80 | 1.33 | −.08 | −1.36 | .176 |
| NS[*] | **.13** | **.06** | **.12** | **2.08** | **.038** |
| HA | −.09 | .07 | −.10 | −1.24 | .215 |
| RD | .11 | .06 | .10 | 1.76 | .079 |
| PS[*] | **.15** | **.07** | **.15** | **2.15** | **.033** |
| SD[*] | **.20** | **.09** | **.21** | **2.31** | **.022** |
| CO[**] | **.24** | **.08** | **.19** | **3.09** | **.002** |
| ST[***] | **.22** | **.06** | **.19** | **3.49** | **<.001** |
| | $F(9, 237) = 18.84, p < .001, R^2 = .42 \ (adj.R^2 = .40)$ | | | | |
| mCERQ total score | | | | | |
| age | −.01 | .28 | −.00 | −0.03 | .98 |
| sex | −1.73 | 1.12 | −.09 | −1.55 | .12 |
| NS | .01 | .05 | .02 | 0.24 | .81 |
| HA[**] | **.19** | **.06** | **.28** | **3.15** | **.00** |
| RD | .09 | .05 | .11 | 1.77 | .08 |
| PS[***] | **.26** | **.06** | **.32** | **4.43** | **<.001** |
| SD[***] | **−.27** | **.07** | **−.36** | **−3.66** | **<.001** |
| CO[*] | **−.14** | **.07** | **−.15** | **−2.19** | **.03** |
| ST | .10 | .05 | .12 | 1.96 | .05 |
| | $F(9, 237) = 12.59, p < .001, R^2 = .32 \ (adj.R^2 = .30)$ | | | | |

**Notes.**

[*]$p < .05$
[**]$p < .01$
[***]$p < .001$

CERQ, Cognitive Emotion Regulation Questionnaire; aCERQ, adaptive CERQ; mCERQ, maladaptive CERQ; TCI-RS, Temperament and Character Inventory-Revised Short; NS, Novelty-Seeking; HA, Harm-Avoidance; RD, Reward-Dependence; PS, Persistence; SD, Self-Directedness; CO, Cooperativeness; ST, Self-Transcendence.

Two-step hierarchical multiple regression analyses were conducted to find how TCI-RS temperament and character explains adaptive and maladaptive strategy scores of CERQ. Sex and age were introduced in the first step as covariates (model 1) and all seven TCI-RS subscales were added in the second step (model 2).

**Table 4  Information criterion for 2-5 latent classes of the nine subscales of the CERQ.**

| Model | BIC | adj. BIC | VLMR $p$ | LMR $p$ | BLRT $p$ | Entropy |
|---|---|---|---|---|---|---|
| 2 class | 11353.27 | 11258.17 | .025 | .026 | .000 | .73 |
| 3 class | 11281.05 | 11154.25 | .694 | .697 | .000 | .78 |
| **4 class** | **11212.21** | **11053.71** | **.009** | **.010** | **.000** | **.82** |
| 5 class | 11216.99 | 11026.79 | .468 | .475 | .000 | .82 |

**Notes.**

LCA, Latent Class Analysis; adj, adjusted; BIC, Bayesian Information Criterion; BLRT, Bootstrapped Likelihood Ratio Test; CERQ, Cognitive Emotion Regulation Questionnaire; LMR, Lo–Mendell–Rubin Likelihood Ratio Test; VLMR, Vuong-Lo-Mendell-Rubin Likelihood Ratio Test.

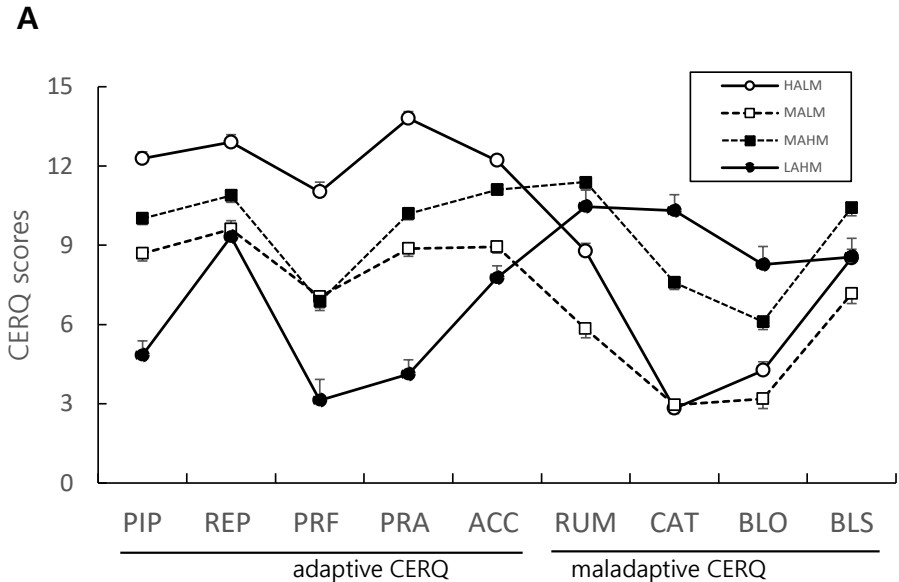

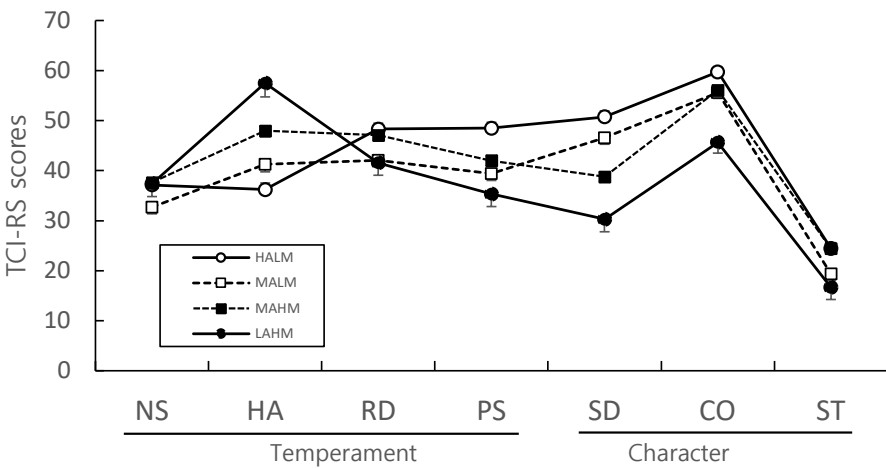

**Figure 1** **(A) Four latent classes in adaptive and maladaptive CERQ subscales. (B) TCI profiles of four latent CERQ classes.** CERQ, Cognitive Emotion Regulation Questionnaire; aCERQ, adaptive CERQ; mCERQ, maladaptive CERQ; TCI-RS, Temperament and Character Inventory-Revised Short; PIP, Putting into perspective; REP, Refocus on planning; PRF, Positive Refocusing; PRA, Positive Reappraisal; ACC, Acceptance; RUM, Rumination; CAT, Catastrophizing; BLO, Blaming Other; BLS, Blaming Self; NS, Novelty-Seeking; HA, Harm-Avoidance; RD, Reward-Dependence; PS, Persistence; SD, Self-Directedness; CO, Cooperativeness; ST, Self-Transcendence. High adaptive and low maladaptive group (HALM, 32.58%) is shown as solid line with white circle, Middle adaptive and low maladaptive group (MALM, 24.70%) with dotted line with white box, Middle adaptive and high maladaptive group (MAHM, 36.84%) with dotted line with black box, Low adaptive and high maladaptive group (LAHM, 6.88%) with solid line and black circle. Data shown as estimated mean and standard error.

to PIP ($F = 63.04$, $p < .001$) and PRA ($F = 108.69$, $p < .001$) scores. REP score was significantly different ($F = 22.58$, $p < .001$) across the classes except between LAHM and MALM, and LAHM and MAHM. PRF score was significantly different ($F = 38.73$, $p < .001$) across the classes except between MALM and MAHM. ACC score was significantly different ($F = 48.80$, $p < .001$) across the classes except between MALM and LAHM. In aCERQ subscales, the four latent classes were significantly different from each other in PIP and PRA scores. REP score was significantly different across the classes except between LAHM and MALM, and LAHM and MAHM. PRF score was significantly different across the classes except between MALM and MAHM. ACC score was significantly different across the classes except between MALM and LAHM.

### Profile analysis of TCI-RS scores and the four latent CERQ classes

Profile Analysis was used to examine significant differences in TCI-RS subscales among the four latent CERQ classes (Fig. 1B). The TCI-RS profiles of the four latent classes were not flat (Greenhouse-Geisser correction, $df = 4.34$, $F = 2.90$, $p < .05$) and the interaction between the four latent classes was significantly different in regard to parallelism (Greenhouse-Geisser correction, $df = 13.02$, $F = 13.14$, $p < .001$), meaning that the four classes were significantly different from each other in terms of all the seven trait mean scores of TCI-RS.

Significant differences in all seven TCI-RS subscales among the four latent classes were found when ANCOVA and Bonferroni post-hoc analysis were performed. In the temperament domain, NS score was significantly different between MALM and MAHM, and HA scores were significantly different from each other except between HALM and MALM. RD scores were significantly different only between MALM and MAHM, and between MALM and HALM. PS scores were significantly different between MALM and the other three latent classes.

In the character domain, SD scores were significantly different across the classes except between MALM and HALM, and CO scores were significantly different from each other except between MALM and MAHM. ST scores were significantly different across the classes except between MALM and LAHM, and between MAHM and HALM.

## DISCUSSION

This study investigated the effects of personality as individual traits as well as complex profiles on cognitive emotion regulation strategies using the TCI-RS and CERQ in a sample of Korean university students. Based on an exploration of latent classes depending on adaptive and maladaptive cognitive emotion regulation, we provide further support that character traits of higher scores in both SD and CO play a pivotal role in predicting psychological adaptation and adjustment.

First, as for the individual personality traits, the temperament trait HA was associated with maladaptive emotion regulation as measured by the CERQ (mCERQ) as in previous studies (*Aldao, Nolen-Hoeksema & Schweizer, 2010*; *Ball, Smolin & Shekhar, 2002*; *Izadpanah et al., 2016*; *Manfredi et al., 2011*; *Peirson & Heuchert, 2001*; *Schreiber, Grant & Odlaug, 2012*). That is, the present results found that temperament traits may guide the direction of cognitive emotion regulation strategy in adaptive or maladaptive ways, with

high scores in HA being related to internalizing problems such as depression, anxiety, worry and catastrophization (*Kim & Lee, 2018*).

In addition, the temperament trait NS was found to be associated with adaptive strategies in this study. This may be expected given that an individual with a high score in NS typically seeks out and enjoys new or novel experiences, ideas and innovative activities. Extraversion and Openness of the five factor model (*McCrae & John, 1992*) which share a similar conceptual basis with NS are known to be correlated with the use of adaptive emotion regulation strategies (*Aluja & Blanch, 2011*; *Hassani et al., 2008*; *Kokkonen & Pulkkinen, 2001*; *Purnamaningsih, 2017*; *Sadr, 2016*; *Sohn & Yoo, 2011*). However, one of the four sub-dimensions of the NS temperament is impulsivity and impulsivity is typically not associated with adaptive emotion regulation (*Schreiber, Grant & Odlaug, 2012*). The association between impulsivity and adaptive emotion regulation should be further explored in future studies.

Second, the temperament trait PS showed a dualistic effect (*Cloninger et al., 2012*), concurrently strengthening both adaptive and maladaptive cognitive emotion regulation strategies. More specifically, PS demonstrated dualistic and opposite effects on cognitive emotion regulation depending on the particular adaptive or maladaptive strategy. This suggests that PS may sustain the employment of emotion regulation strategy in both adaptive as well as maladaptive ways.

PS is a tendency to maintain one's behavior even in the face of punishment or compensation, and individuals with higher PS can be characterized as industrious, determined, ambitious and perfectionistic (*Cloninger, 1987*; *Cloninger, Svrakic & Przybeck, 1993*). An individual with high scores in HA and PS may tend to use maladaptive cognitive emotion regulation strategies since an individual with high scores in HA may focus on the dangers and worries related to the situation and employ pessimistic interpretation and avoidant responses. Therefore, they may be more sensitive to criticism and the possibility of punishment, which may further reinforce rigid maintenance of their cognitive style. However, an individual with high scores in NS and PS tended to use adaptive cognitive emotion regulation strategies. High scores in NS prevent loss of motivation resulting from non-reward or frustration despite the difficulty of the situation and this may make the person withstand and sustain their current cognitive style in a more adaptive way.

Third, it was also found that a high score in the character trait of SD promotes the use of adaptive emotion regulation and suppresses the use of maladaptive emotion regulation strategy in addition to the temperament traits such as higher HA and PS. According to Cloninger (*Josefsson et al., 2013*), SD is the first character trait that develops among humans and serves as the foundation of character development (*Josefsson et al., 2013*). Those who score high in the character trait of SD are considered as responsible, purposeful, resourceful, self-accepting, and disciplined, suggesting that such individuals regard themselves as autonomous (*Cloninger, Svrakic & Przybeck, 1993*). Therefore, high scores in SD are thought to serve as a protective factor in moderating unhealthy ways of thinking and encouraging the use of more well-functioning means of emotion regulation (*Cloninger & Zohar, 2011*; *Josefsson et al., 2011*).

When person-centered analyses were performed on the complex nature of temperament and character using LCA and Profile Analysis, there were four latent CERQ classes showing distinct personality classes. Individuals with a highly adaptive emotion regulation style/profile (i.e., HALM) exhibited the highest levels of responsibility, resourcefulness, and self-acceptance (i.e., high scores in SD) and also the highest levels of optimism and energy levels (i.e., low scores in HA). In contrast, individuals with a maladaptive emotion regulation style/profile (i.e., LAHM) were characterized by low self-esteem, feelings of victimization, and lack of goals (i.e., low scores in SD) and also by pessimism, low energy, and propensity for anxiety (i.e., high scores in HA) (*Cloninger, 1987*; *Cloninger, Svrakic & Przybeck, 1993*). Indeed, resilience has been proposed as the result of this specific personality configuration (*Cloninger, 2004*; *Eley et al., 2013*). That being said, resilience or adaptive functioning has also been associated with industriousness, perseverance, and hard-working style (i.e., high scores in PS). Thus, one remaining question is why individuals with a highly adaptive emotion regulation style/profile did not demonstrate higher scores in PS compared to individuals with a maladaptive emotion regulation style/profile. It may be that high levels of PS are only adaptive in combination with high scores in SD and low scores in HA. In other words, high levels of PS are also a feature of individuals who may exhibit high HA and low SD. This personality profile, however, is vulnerable for burn out. Hence, it is possible that the lack of difference in PS between individuals in the HALM and MALM groups is due to the fact that PS enables the maintenance of either adaptive or maladaptive emotion regulation strategies depending on how prone the individual is to worry (i.e., HA) and self-directedness (*Cloninger et al., 2012*). That is, the findings regarding the combination of high score in SD with low score in HA are consistent with previous findings that support the idea of psychological maturity or healthy functioning self (*Josefsson et al., 2013*), indicating that individuals may appropriately regulate their emotions depending on their circumstances.

Fourth, another important new finding in our study was that high scores in SD and high scores in CO and/or low scores in ST may play an important role in fast and adaptive emotional response or in the choice of cognitive emotion regulation strategy in stress situations which has not been previously reported. The importance of SD and CO in our stress response has been consistently reported (*Eley et al., 2013*; *Eley et al., 2016*; *Kim, Lee & Lee, 2013*; *Lee, Choi & Chae, 2017*; *Sievert et al., 2016*), and the combination of SD and CO scores representing psychological maturity was found to be an objective factor related to degree of social support, life satisfaction, subjective well-being, composite health index, and happiness (*Eley et al., 2013*). In addition, the combination of high scores in ST with low scores in SD is associated with maladaptive functioning and is an indicator of an immature character profile (*Lee, Choi & Chae, 2017*; *Spittlehouse et al., 2014*). That is, high scores in ST in combination with high scores in SD are advantageous in that this profile reflects creativity, holism, and contemplation but high scores in ST in combination with low scores in SD are disadvantageous as this may represent magical and unrealistic thinking. Once again, high scores in SD appear to be a prerequisite for advantages offered by high scores in ST (*Cloninger & Zohar, 2011*; *Josefsson et al., 2011*; *Moreira et al., 2015*).

While extant studies primarily focused on the role of temperament, the present study implicates the significant role of character in emotion regulation. The character domain of TCI-RS measures psychological maturity and is reported to be a protective factor in the development of psychopathology and is also related to resilience to stress and biopsychological well-being (*Eley et al., 2013*; *Moreira et al., 2015*). This study suggests that cognitive emotion regulation may be a mechanism that potentially explains the effects of character on psychological health. Furthermore, character may reflect an individual's cognitive style which promotes reappraisal of a given environmental stimulus as a task to be solved, not as a dangerous threat, and thus prepare their behavioral response accordingly (*Lee, Choi & Chae, 2017*).

As such, temperament and character, which play a pivotal role in well-being and mental health, affect long-term development and continuation of emotion regulation style to everyday life stressors (*Garnefski, Kraaij & Spinhoven, 2001*), not only on an individual trait level but also through a complex interaction (*Eley et al., 2016*; *Lee, Choi & Chae, 2017*). In this way, person-centered analysis such as LCA or Profile Analysis on psychological characteristics is a great advantage of this study. The LCA and Profile Analysis (*Eley et al., 2016*) provide better clinical explanation compared to cluster analysis (*Kim, 2008*; *Sievert et al., 2016*) in which mature and adaptive strategy in relation to stress emerges through a complex and non-linear interaction (*Cloninger et al., 2012*). This study also analyzed the complex interaction of temperament and character traits in maladaptive and adaptive emotion regulation strategies which was not investigated before.

There are a number of limitations of this study. First, this study was conducted with a relatively small sample of Korean university students. The findings should be supported using a bigger sample balanced in sex and age. Second, the sex differences of Korean university students in personality and cognitive emotion regulation strategy should be further examined to determine whether cross-cultural differences (*Ahn, Lee & Joo, 2013*; *Cho et al., 2007*) exist in general.

This study examined maladaptive and adaptive cognitive emotion regulation strategies using the two perspectives of individual traits with two-step hierarchical regression analyses and the complex interaction across the classes simultaneously through LCA. The temperament domain guided the direction of emotion regulation such that high scores in HA and PS predicted maladaptive cognitive emotion regulation strategy and high scores in NS and PS predicted adaptive strategy. As for the character domain, SD and CO were correlated positively with adaptive and negatively with maladaptive cognitive emotion regulation strategy which may explain the significance of resilience to stress and promotion of well-being.

## CONCLUSION

Specific combination of temperament and character traits appears to be particularly influential in predicting adaptive and maladaptive emotion regulation strategies and is also associated with both mental health and well-being. Mature personality as represented by character traits such as SD and CO rather than temperament traits was found again to

account for temperamental vulnerability as well as specific cognitive emotion regulation. In particular, the effect of PS was once again found to be dualistic, supporting the concept of a non-linear model. In other words, the same high score in PS may exert an opposite tendency. It may be associated with adaptive emotion regulation when combined with high scores in NS but with maladaptive emotion regulation when combined with high scores in HA. Since adaptive emotion regulation is pivotal for psychological health, structured education and persistent interest on character development in Korean university students should be emphasized. This study provides a foundation for a potential psychological mechanism in stress response as well as existing psychopathological studies.

### Funding

This research was supported by Kyungsung University Research Grants in 2018. The funders had no role in study design, data collection and analysis, decision to publish, or preparation of the manuscript.

### Grant Disclosures

The following grant information was disclosed by the authors:
Kyungsung University Research.

### Competing Interests

The authors declare there are no competing interests.

### Author Contributions

- Han Chae and Soo Jin Lee conceived and designed the experiments, performed the experiments, analyzed the data, contributed reagents/materials/analysis tools, prepared figures and/or tables, approved the final draft.
- Soo Hyun Park and Danilo Garcia conceived and designed the experiments, analyzed the data, authored or reviewed drafts of the paper, approved the final draft.

### Human Ethics

The following information was supplied relating to ethical approvals (i.e., approving body and any reference numbers):

The institutional ethic board of Pusan National University in South Korea granted ethical approval to carry out the study within its facilities (PNU IRB/2015_59_HR).

### Data Availability

The raw data and code are available in the Supplemental Files.

### Supplemental Information

Supplemental information for this article can be found online at http://dx.doi.org/10.7717/peerj.7958#supplemental-information.

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
