# Peer review of "Cloninger’s TCI associations with adaptive and maladaptive emotion regulation strategies"

_PeerJ, doi:10.7717/peerj.7958_

## Round 0.1 · original submission · Major Revisions

Reviewer 1 provides specific and thoughtful suggestions for improving the manuscript. Please address each of them.

The request to delete lines in the table is less crucial but may facilitate reading for the reviewer.

Reviewer 1 ·

Basic reporting

Use “participants” not “subjects”.
Use “sex” not “gender” (they are not interchangeable).
Explain abbreviations at the first use only, and then use the abbreviation throughout the manuscript.
Confirm which language the CERQ questionnaires is in and reference the translation papers for both the CERQ and TCI.
Throughout the document the number of decimal places changes from 1 to 2 and sometimes to 3. Please be consistent (see line 151 compared to line 153). Also, leading zeros are sometimes used but not always.
The authors could replace “TCI” with “TCI-RS” when referring to the version used for this study because the TCI-RS is 100 items shorter than the TCI-R and the reader needs to be clear that the shorter version is being used.
The authors don’t mention that there are advantages of both high and low temperament scores but, for character, only high scores are adaptive.
Throughout the manuscript the use of confusing terms such as personality domains, personality traits, personality subscales and personality profiles make the manuscript hard to follow. The authors began by explaining that Cloninger’s personality model has 2 domains: temperament and character. The temperament domain has four traits (RD, HA, NS and PS) and character has three (SD, CO and ST). Then at line 201 the authors refer to “TCI subscale profile”, which is confusing. Generally speaking, personality “profiles” refer to high and low combinations of personality traits (see Spittlehouse et al., 2014 for an example of character profiles). There is a similar problem with the latent classes or groups. For example, sometimes the authors refer to the latent classes as subgroups (see line 299) which is unclear. Please correct this throughout the manuscript.
The results section is untidy. The authors may want to consider putting in either the t value or the p value (but not both), or only quoting the highly relevant statistics.

Tables:
There is no indication in the manuscript of where the figures and tables should be placed.
Remove all the horizontal lines except the top and bottom ones.
Some p values, correlations and other values are shown with leading zeros and some without. Please be consistent.
It is unclear what is being shown in the tables. For example in Table 1 age is shown as the mean and standard deviation but this is not explicit. Further down the table are the CERQ scores are presumably shown as means and standard deviations but again it is not clear. Please label the tables appropriately.
Use abbreviations in the table (e.g. CERQ in table one) and write it out in full in the legend/footnote (as has been done in table 3).
Some tables show *** to indicate the strength of p values and have the p value listed too. The tables would be clearer if one or the other are shown. It may be best to show the p value and bold the values that are significant.
Align all columns (see the p value column in table 1).
Be consistent with the number of decimal places within a table.
The titles of tables 2, 3 and 4 are poorly written and not informative.
Figures:
The Y axes need labelling (i.e. figure 1A Y axis: CERQ scores)
The overall and individual figure titles are not informative (e.g figure 1A: four latent classes in CERQ nine subscales?).
The abbreviations do not need to be written out in full. Use the footnote to explain abbreviations.
The X axes would benefit from an overarching bracket showing:
figure 1A: the adaptive CERQ (PIP, REP, PRF, PRA, ACC) and another one for the maladaptive CERQ (RUM, CAT, BLO, BLS).
Figure 1B: TCI-RS Temperament (NS, HA, RD PS) and character (SD, CO, ST).
I don’t think the paragraph below the figures adds anything because explanations of the different lines are shown in the legend and the percentages of the four latent classes are described in the results section.

Experimental design

The methods, in particular the statistical analysis section, need to be considerably revised. As it stands it is hard to follow and the analysis cannot be replicated because of the lack of details and clarity. For example, what type of multiple regression was used (stepwise is mentioned in the discussion but not in the statistical analysis section) and which variables were entered at each step? Why was stepwise regression used? Also, Bonferroni post-hoc analysis is mentioned in the results (line 263) but not in the statistical analysis section. Specific comments are in the annotated PDF.

Latent classes: Why was the four class model chosen? Which fit statistics were most important? This is not explicit.

Validity of the findings

Discussion:
The discussion has the following problems which make it unclear, difficult to follow and hard to assess the validity of the findings:
General lack of clear language or grammar errors e.g. line 306 traits not trait. Line 310 “the temperament NS” should be “the temperament trait NS”. Line 311 “that an individual with high NS score” would be better as “that an individual with a high NS score” etc.
Inappropriate language: line 312 passion? Hidden rewards? Lines 323 and 325 “positive scores” should be “high scores”. Line 332 stimulates? Line 357 prompt? Line 364 well- functioning? Etc.
Missing references: 5 factor model on line 313, line 334 According to Cloninger.
Inappropriate references: line 347 Lee at al., what is this referencing? Line 358 too.
Long sentences that are hard to follow, for example see lines 324-328.
Repetition of results without critical interpretation for example, see lines 344-351.
Lack of specifics: line 329 “since NS prevents loss of motivation”. Presumably the authors mean high NS? Likewise, line 332 High “SD stimulates the…”. Line 356-357 “the character domains of SD, CO and ST may…”, High or low SD, CO and ST?
Line 336-339 may be better in the methods section.
The authors do not mention the lack of contribution from the character trait ST. I think that, of the three character traits, being high on ST does not appear to advantageous. This is worth discussion.

Data file:
Needs metadata (for example, which are male 1 or 2?)
The authors could consider providing the statistical code/syntax used, especially for the Mplus latent class analysis.

Additional comments

The authors have identified a knowledge gap, that personality and emotion regulation have not been sufficiently examined, and set the scene reasonably well in the introduction. However, the methods section does not provide enough information to replicate the study and the discussion suffers from poor use of English making it hard to read and interpret. With major revisions this paper may be suitable for publication but would require a further review.

Annotated reviews are not available for download in order to protect the identity of reviewers who chose to remain anonymous.

·

Basic reporting

I found this paper very easy to read even though it described complex statistics and research design. It was written clearly in its rational through to the final message of the paper. The literature cited was appropriate and comprehensive to cover the research reported. The tables and figures were all clear and well described.

Experimental design

This is an important paper and adds to the literature around personality and its influence on our coping in everyday life. Emotion regulation was studied through the use of latent class analysis and then relating it to temperament and character. The design is appropriate and well performed. Furthermore, the methods were clear and thorough. The use of LCA in producing profiles of was a good way to illustrate the relationship between temperament and character traits and emotion regulation.

Validity of the findings

This study contributes new information about how certain temperament and character traits are related to cognitive emotion regulation. The findings that Harm Avoidance and Self-Directedness are highly influential to emotion regulation adds to what we know about their role in mental well-being. The authors conclusions sum up their findings clearly.

Additional comments

I enjoyed reading this paper. It makes an important contribution to our understanding of mental well-being and psychological health in general.

---

## Round 0.2 · Minor Revisions

Please address each of the reviewer's concerns.

Reviewer 1 ·

Basic reporting

Although this manuscript is much improved from that last review there are still considerable problems with the use of English. The discussion needs particular attention. There are many examples, here are a few:
Line 80-82 sentence does not make sense.
Line 211-212 sentence does not make sense.
Line 250-252 sentence is poorly written.
Line 298-303 sentence is poorly written. Significant differences in what? TCI scores?
Line 309-317 is poorly written. E.g. …PS score was significantly between MALM and the other three latent classes…??
Line 405-409 sentence does not make sense.
Line 170-172 describes the character aspects of Cloninger’s TCI. It was mentioned in the previous review that the authors need to explicitly state whether they are talking about high or low scores on a personality trait. For example low scores on the trait SD are not characterised by being purposeful, responsible etc. but, in this sentence (and many others), the authors do not state that the traits described are seen in those with high scores. Although the authors have corrected some of these errors pointed out in the last review, they have not systematically gone through the manuscript and corrected all of these errors. These need to be done thoroughly.

Minor grammatical errors appear throughout. Here are a few from the introduction:
Line 33: PS were pivotal for adaptive….?
Line 71 …those high in neuroticism…..
Line 102-103 …while higher HA… should be while higher scores in HA. The problem appears in the sentence on lines 106-108.
Other (minor issues):
Using numerals rather than words for numbers 1-9. See line 142, 187, 303 etc.
In several places the authors use TCI profile (see line 207, 297 etc) when they mean TCI subscale. They have not created TCI profiles rather, they are comparing scores on TCI subscales across the four latent classes. I mentioned this in my previous review but the authors have not addressed it sufficiently.

Generally speaking the rule to follow regarding leading zeros is:
• If a value has the potential to exceed 1.0, use the leading zero.
• If a value can never exceed 1.0, do not use the leading zero.
Please adjust the manuscript accordingly, paying particular attention to the t-test values starting at line 220.
Lines 144-157: Examples of questions from the CERQ would be better in quote marks.
Internal consistencies for the present study are given for the CERQ but not the TCI. Please include them for the TCI.
Line 187 first use of “LCA” should be written in full with the abbreviation in brackets.
Please include the version of MPlus used in line 188.
Line 214 …p values of .05, 0.1, etc…. presumably should be p values <.05 etc?
Insert “the” before the word “regression” on line 253 and 255 and also before “variance” on line 256.
Line 262 title would be better as “LCA of CERQ subscale scores”.
Line 296 title would be more accurate as “Profile analysis of TCI-RS scores and the four latent CERQ classes”.
Line 266 give the exact entropy value.
Table 4: Something like “Information criterion for 2-5 latent classes of the nine subscales of the CERQ” would be a more informative title.
The paper title may be more accurate as “Cloninger’s TCI associations with adaptive and maladaptive emotion regulation strategies”.
References: In the text please distinguish between papers by the same author in the same year. E.g. Lee et al., 2017 (line 396) there are two papers by Lee in 2017.

Experimental design

Experimental design is fine there is now sufficient information to replicate the study.
The structure and content of the introduction are fine except the aim at the end of the introduction is vague and misses out some of the analysis and results (e.g. the multiple regression). Something like the following would be better:
The aims of this study were to explore the association of personality on adaptive and maladaptive cognitive emotion regulation by (1) assessing how much variance in emotion regulation scores can be explained by personality scores and (2) by identifying groups of emotion regulation style and assessing differences in personality scores between these groups etc.

Validity of the findings

Discussion:
Structure: The first paragraph (lines 321-325) would be improved by including the main personality findings from the study. Which TCI traits had the most impact on emotion regulation?
The rest of discussion may benefit from being reconstructed so that the most important variables are discussed first, in order to highlight the significant findings, i.e. discussing SD, CO then PS rather than HA, NS then PS.
The discussion about NS does not include that high scores on NS are associated with impulsivity. Impulsivity is not usually associated with adaptive emotion regulation (see Schreiber et. Al. 2012). This should be discussed.
There are a few statements in the discussion that are not referenced (for example: the sentence on lines 348-350). Please correct this.
Self-transcendence (Lines 395-400): High SD is not always necessary for high ST to have a positive effect and high ST is not necessarily advantageous if SD and CO are high (Spittlehouse et al., 2014). Please adjust the statement accordingly. Also, the lee et al. 2017 reference appears to be the wrong citation.
The conclusion doesn’t match what is written in the abstract i.e. there is no mention of the influence of HA in the conclusion.
References
Liana R.N. Schreiber, Jon E. Grant, Brian L. Odlaug (2012). Emotion Regulation and Impulsivity in Young Adults. J Psychiatr Res. May; 46(5): 651–658. doi: 10.1016/j.jpsychires.2012.02.005.

Spittlehouse, J. K., Vierck, E., Pearson, J. F. & Joyce, P. R. (2014). Temperament and character as determinants of well-being. Comprehensive psychiatry 55, 1679-1687.

Additional comments

As mentioned before the authors have identified a knowledge gap, that personality and emotion regulation have not been sufficiently examined, and they set the scene well in the introduction. The methods section is much improved and now provides enough information to replicate the study. The results, tables and figures have been revised adequately. The discussion has also been revised but poor use of English is still a problem making some parts difficult to follow. I recommend major revisions (mainly around the use of English) are still required before the paper is suitable for publication.

---

## Round 0.3 · Minor Revisions

1. Please cite Spittlehouse et al as suggested by the previous reviewer, or explain why not

2. line 130: how were participants selected from the total population of college students? Did some choose to respond to the survey and some not? If so, what was the response rate? Were they required to do the survey to get a grade in a class? etc.

3. The English still needs revision. I made notes reading through that may help, but you'll need to have a native speaker proofread the rest of the text carefully, too. Here are my notes:

ABSTRACT
the characters Self-Directedness (SD) and Cooperativeness (CO)
-> the character traits S....

The temperament HA and character SD were
(ditto)

This study demonstrated that SD and CO are related to cognitive emotion regulation strategies along with psychological health and well-being, and the dualistic effects of PS when combined with NS or HA on response to stressful situations.
-> Not parallel (clause "that SD and CO are related ...", followed by noun phrase "the dualistic effects")

matured character
-> mature character

The importance of developing matured character ... were discussed.
verb number

l. 60 Such two separate strategies ...
-> Two separate strategies ...

passim: "matured" in place of "mature"

l. 104: scores ... is
-> scores ... are

l. 204: 2) the significant probability value
-> 2) the more significant the probability value

ll. 206-207: rewrite # (3)

ll. 210-211: Bonferroni post-hoc analysis was performed to confirm where the differences occurred among them.
(?? please clarify)

l. 219: "and p values of <.05, <.01, etc. were used for significance."
-> what does this mean? Usually a sentence like this is intended to clarify that p<.05 (for instance) was chosen a priori as the threshold for significance. If that's not what you did, omit this phrase

ll. 270-271: please correct definite and indefinite pronoun usage

l. 295 four latent classes
-> the four latent classes

ll. 298-299 "from each other" rewrite this

l. 348 in the future studies
-> in future studies

ll. 356-357 An individual ... tended to use maladaptive
-> An individual ... may tend to use a maladaptive

(I gave up at this point, but there are several other errors in this paragraph and it gets worse beyond this point--please continue to edit carefully)

---

## Round 0.4 · Minor Revisions

Very minor changes needed.

I'm still not clear on my 2nd point from the previous submission. I think you're saying that students in a given class were asked to participate and most but not all did. You can email me, if you want, to clarify.

The English language quality is much improved but there are still some errors. Examples:
= line 104 "higher scores in SD is"
= 132 "The CERQ and TCI-RS were inquired"
= 203 "Baysian" should read "the Bayesian"
= line 358 "scores ... prevents"
= line 365 "SD is the first to be achieved than any other character traits"
= line 366 "the character SD"
= 406-407 typos in citation
= 449 "Particular combination of temperament and character traits appears"

---

## Round 0.5 · accepted · Accept

Congratulations on the Acceptance